# In Vitro Fecal Fermentation of High Pressure-Treated Fruit Peels Used as Dietary Fiber Sources

**DOI:** 10.3390/molecules24040697

**Published:** 2019-02-15

**Authors:** Viridiana Tejada-Ortigoza, Luis Eduardo Garcia-Amezquita, Ahmad E. Kazem, Osvaldo H. Campanella, M. Pilar Cano, Bruce R. Hamaker, Sergio O. Serna-Saldívar, Jorge Welti-Chanes

**Affiliations:** 1Tecnológico de Monterrey, Escuela de Ingeniería y Ciencias, Epigmenio González 500, Santiago de Querétaro, QRO 76130, Mexico; viri.tejada@tec.mx; 2Tecnológico de Monterrey, Escuela de Ingeniería y Ciencias, General Ramón Corona 2514, Zapopan, JC 45138, Mexico; garcia.amezquita@tec.mx; 3Whistler Center of Carbohydrate Research, Food Science Department, Purdue University, 745 Agricultural Mall Drive, West Lafayette, IN, 47907, USA; akazem@purdue.edu (A.E.K.); campanella.20@osu.edu (O.H.C.); hamakerb@purdue.edu (B.R.H.); 4Department of Food Science and Technology, 110 Parker Food Science Building, The Ohio State University, 2015 Fyffe Road, Columbus, OH, 43210-1007, USA; 5Tecnológico de Monterrey, Escuela de Ingeniería y Ciencias, Centro de Biotecnología FEMSA, Eugenio Garza Sada 2501, Monterrey, NL 64849, Mexico; mpilar.cano@csic.es (M.P.C.); sserna@tec.mx (S.O.S.-S.); 6Department of Biotechnology and Food Microbiology, Institute of Food Science Research (CIAL) (CSIC-UAM), C/Nicolás Cabrera 9, 28049 Madrid, Spain

**Keywords:** high hydrostatic pressure, fruit peel dietary fiber, short chain fatty acid, fecal fermentation

## Abstract

Fruit by-products are being investigated as non-conventional alternative sources of dietary fiber (DF). High hydrostatic pressure (HHP) treatments have been used to modify DF content as well as its technological and physiological functionality. Orange, mango and prickly pear peels untreated (OU, MU and PPU) and HHP-treated at 600 MPa (OP/55 °C and 20 min, MP/22 °C and 10 min, PPP/55 °C and 10 min) were evaluated. Untreated and treated fruit peels were subjected to fecal in vitro fermentations. The neutral sugar composition and linkage glycosidic positions were related to the production of short chain fatty acids (SCFA) resulting from the fermentation of the materials. After HHP-treatments, changes from multibranched sugars to linear sugars were observed. After 24 h of fermentation, OP yielded the highest amount of SCFA followed by PPU and MP (389.4, 282.0 and 204.6 μmol/10 mg DF, respectively). HHP treatment increased the SCFA concentration of orange and mango peel by 7 and 10.3% respectively, compared with the untreated samples after 24 h of fermentation. The results presented herein suggest that fruit peels could be used as good fermentable fiber sources, because they yielded high amounts of SCFA during in vitro fermentations.

## 1. Introduction

The role of dietary fiber (DF) in human health is highly recognized and associated with satiety and attenuation of constipation, diminishment of both glucose and lipid levels, reduced risk of coronary heart disease and cancer and enhancement of the growth of beneficial of hind gut microbiota [1,2]. The human gut microbiome is made of different microbial strains with special physiological implications in health and wellness [3]. The importance of DF and colonic bacteria relies on the fermentation processes and products occurring in the gut, mainly the production of short chain fatty acids (SCFA) such as propionate, acetate and butyrate. Among other benefits, SCFAs improve the absorption of minerals, reduce the production of bile acids and inhibit the growth of pathogenic bacteria [4,5,6]. Acetate is the chief SCFA in the colon—it gets into the peripheral circulation and, once absorbed, increases cholesterol synthesis. Propionate is used as substrate for gluconeogenesis in the liver, and it has been reported to inhibit endogenous cholesterol synthesis. A reduction in the acetate/propionate ratio is related to the diminishment of serum lipids and risk of cardiovascular diseases. Butyrate is used by colonocytes as the major source of energy and it has an important participation in sustaining the colonic mucosa and in preventing colon cancer. Butyrate promotes cell differentiation and proliferation as well as apoptosis of colonocytes by transforming the primary to secondary bile acids by colonic acidification [6].

The fermentation rate highly depends on the DF structure, which is mainly associated with the chemical composition and types of linkages. In general, insoluble dietary fiber (IDF) passes through the colon either intact, contributing to stool bulking, or slowly fermenting, whereas soluble dietary fiber (SDF) and oligosaccharides readily ferment in the cecum and proximal colon, typically with little carbohydrate fermentation in the distal region of the colon. The above often leads to a high concentration of SCFAs produced by fermentation in the proximal colon, and a low concentration in the distal part of the colon [7]. Additionally, the high fermentation rate of SDF can cause bloating and discomfort due to the production of gases, with a prolonged fermentation profile preferable [8]. Thus, not only IDF, but other DF compositions and structures are capable of slowing the SCFA fermentation rate. Several studies have reported that simple structures, different molecular weights, the degree of substitution and certain types of linkages affect SCFA production and rates [2,7,9,10,11].

While cereal grains and their by-products are the most consumed sources of DF, fruit and vegetable by-products are being investigated as non-conventional alternative sources due to their high DF contents and different SDF/IDF ratios. Some authors have reported that total DF content in fruit and vegetable by-products can be from 40–90% dry weight (dw), depending on the source and extraction. Particularly, fruit peels have important amount of pectins, cellulose, hemicellulose, gums, mucilages and some other polysaccharides that makes them a good source of DF [12,13].

Chemical, physical and thermal food differently affect DF properties such as the SDF/IDF ratio and functionality [14]. High hydrostatic pressure (HHP) treatments have been used as an alternative to modify DF content, increasing the SDF content, and to modify its technological functionality with less severe processing conditions [15,16]. Tejada-Ortigoza et al. [13] reported that HHP induced DF modifications in mango, orange and prickly pear peels, increasing the SDF and maintaining the total dietary fiber (TDF) content of the materials. These changes considerably impact the DF technological functional properties, such as water and oil holding capacity, swelling capacity, etc. However, the evaluation of their physiological functional properties as a promising fermentation substrate for SCFA production remains unknown.

The main objective of this study was to compare the neutral sugar profile and the sugar linkage proportion of DF in mango, orange and prickly pear peel related to gas production, changes in pH and short chain fatty acid profile after in vitro fecal fermentations. Secondarily, the modifications caused by HHP treatments on these fibers were also evaluated.

## 2. Results and Discussion

### 2.1. Chemical Characterization of Fruit Peels

The SDF and IDF contents of samples expressed as % dry weight were 7.2 and 46.3 (OU), 14.0 and 38.5 (OP); 22.1 and 33.4 (MU), 20.7 and 29.5 (MP); and 6.7 and 30.9 (PPU) and 6.8 and 37.2 (PPP), respectively. These DF values were previously reported by Tejada-Ortigoza et al. [13]. The % SDF/TDF of each sample was 13.7 (OU), 26.9 (OP), 40.1 (MU), 41.1 (MP), 17.7 (PPU) and 15.7 (PPP), indicating a wide variation in this parameter among treated and untreated samples. Neutral sugar profiles of untreated and HHP-treated fruit peels are depicted in Table 1. For all tested fruit peels, the major sugar component was arabinose, followed by galactose and glucose. The high amounts of arabinose and galactose could be related to neutral arabinogalactan polysaccharides or pectic polysaccharides linked to galacturonic acid residues, whereas the high glucose content in these DF fruit peels might be related to the presence of cellulosic cell wall polysaccharides and β-glucans [17]. However, different neutral sugars profile can be observed among the studied fruit peels. Similar profiles were previously reported for mango [17] and prickly pear peels [18].

In general, the sugar profile was slightly changed after the HHP treatment with variations among sugars of ~3.0%. The highest content of xylose in PPU suggests the presence of xylans, which are the most abundant component of hemicelluloses in cell wall plants of this fruit peel [19]. Also, the arabinose associated with this fruit peel is mainly related to the presence of rhamnogalacturonan substituted by arabinans or arabinogalactans [20]. A noticeable reduction in the percentage of arabinose in PPP compared to PPU was observed. This notorious difference has been explained as the transformation of water-insoluble xyloglucans into water-soluble xyloglucans and vice versa [21]. Tejada-Ortigoza et al. [13] reported changes in the DF profile observed after processing, which included an increase of IDF content from 30.9 to 37.2% (dw). In this case, HHP might have caused damage to fiber-rich cell walls, breaking internal structures that promoted mass transfer and thus increasing permeability and the release of polysaccharides that were not available for quantification [21,22]. Some authors have also suggested that higher DF contents are the result of the formation of complexes between polysaccharides and other food components, which are quantified as DF [21].

Thirteen, nine and eleven major sugar linkages were identified in orange (Figure 1A), mango (Figure 1B) and prickly pear peels (Figure 1C), respectively. In the case of orange peel, decreases of up to 4% in the multibranched xylose and glucose (2,3,4-Xyl; 2,3,4-Glc), with an increase of ~10% of 4-Glc were observed after processing. A similar trend was also observed for mango peel, where the proportions of 2,3,4-Xyl (from 27 to 18%) and 4,6-Glc (from 11 to 7%) were reduced and the proportion of 4-Glc increased ~11% after HHP treatment. PPP also presented diminishments from 24 to 17% in the proportion of 2,3,4-Glc. However, in this fruit peel, the 4-Glc linkage was reduced as well, accompanied with slight increases of up to 2.6% of multibranched xylose and glucose (2,3,4-Xyl; 2,6-Glc). The reduction of substituted sugars due to HHP treatments could be related to the cleavage of glycosidic bonds or the rupture of weak unions among polysaccharides. Heating and/or compression are effects caused by HHP, which are known to modify food components such as polysaccharides branching [23]. In arabinoxylans, differences in structural characteristics have shown different susceptibilities to microbiota degradation. It has been reported that highly branched sugars have a lower susceptibility to fermentation in the hind gut [2].

The different neutral sugar composition and linkages found in this study might have physiological implications relying in the selective consumption of DF by specific microbial groups, although to achieve a definitive conclusion further research is needed.

### 2.2. In Vitro Fecal Fermentation of Fruit Peels

#### 2.2.1. Total Gas Production and pH Changes

HHP-treated and untreated mango and orange peel had similar initial fermentation rates (0–6 h) ranging from 79.4 to 89.9 µL gas/mg DF compared to fructo-oligosaccharides (FOS) (83.9 µL gas/mg DF) (Figure 2). The highest fermentation rate achieved during the first 6 h was for orange, both OU and OP, with a gas production of 112 µL/mg DF. During the next 6 h of fermentation (6–12 h), the rate was maintained for mango and prickly pear peels with the lowest value observed for MU and the highest for PPU, 110.9 and 133.2 µL gas/mg DF, respectively. DFs with a low and steady gas production rate might not result in bloating, because the gases would be expelled slowly through the lungs and anus, diminishing the discomfort [24]. In the case of orange and FOS, the samples were rapidly fermented, with rising values of up to 200.0 µL gas/mg DF. After 24 h of fermentation the differences were more noticeable, where FOS exerted the highest gas production (290.0 µL/mg DF), followed by orange, prickly pear (PPU > PPP) and finally mango peel (MP > MU) with a production of about half the amount of FOS (~143.6 µL gas/mg DF). A slight but significant decrease in the production of gas after 24 h of fermentation in PPP was observed. A low initial rate of gas production and a complete fermentation through the colon would be the desirable characteristics of a DF [2]. Based on gas production, both mango peel samples were well fermented during the first stage but the fermentation was constant after 24 h. These incomplete fermentations might be due to the complex structure of polysaccharides, the high degree of branching and the amount of side chain residues [8]. HPP is a treatment that applies significant pressure to the samples but little shear force, therefore these results indicate the need of shear forces to promote more drastic changes on the DF structures. For cereal arabinoxylans (AX), the slowly fermented ones presented a high degree of branching with single xylose units and trisaccharide branch chains. The simpler structures were associated with a rapid initial fermentation rate [2].

Figure 3 depicts the changes in pH associated with the fermentation. The behavior presented was closely related to that observed for gas production. During the first 6 h of fermentation, orange peel samples lowered the pH from an initial ~7.86 to 7.0, while mango samples exhibited the lowest diminishment (7.36–7.45). The pH pattern of orange < prickly pear < mango peel was maintained during the whole fermentation period. The lowest pH was achieved by FOS after 12 h of fermentation with a value of 6.43. Interestingly, no significant pH effects were observed when the untreated and the HHP-treated samples were compared.

#### 2.2.2. SCFA Production and Profiles

As expected, the generation of SCFA matched the observed pH changes for all peel samples (Figure 4). During the first 6 h of fermentation, orange peel generated the highest SCFA content, followed by prickly pear and mango peels. The positive control, FOS, generated a comparatively lower amount of total SCFA (51.2 μmol/10 mg DF) which was similar to mango peel (43.3–60.4 μmol/10 mg DF). A similar behavior of SCFA production among fruit peels (orange > prickly pear > mango) was observed after 12 h of fermentation. At this fermentation time, values of ~310.0 μmol/10 mg DF and 351.3 μmol/10 mg DF were observed for orange peel and FOS, respectively. After 24 h of fermentation, the maximum SCFA production was achieved by FOS (496.0 μmol/10 mg DF), followed again by orange, prickly pear and mango peel samples. Orange and prickly pear samples reached between 362.2 (OU) and 389.4 μmol/10 mg DF (OP) and between 269.8 (MP) and 282.0 μmol/10 mg DF (MU), respectively. No significant differences were observed between the untreated and HHP-treated orange and prickly pear peels, but a significant improvement in the SCFA production was observed with the treatment of mango peel.

Orange peel had the highest IDF content [16], and the fermentation of these samples resulted in the highest SCFA production during the first 24 h of fermentation, just behind FOS. The prickly pear peel samples followed orange peels in IDF content (30.0–37.2% dw), and their fermentation also generated high levels of SCFA. However, even though the HHP treatment changed the SDF/TDF ratio considerably from 13.7 to 26.9% in the case of orange peel, increased the IDF content from 30.9 to 37.2% in prickly pear peel and reduced the substituted sugars (Figure 1), no noticeable changes were observed between treated and untreated samples in regard to SCFA production. Similar results were observed in durum wheat DF, where after a 3-fold increase in the SDF content caused by an enzymatic treatment no effect on the formation of SCFA was observed. However, the increment in SDF stimulated the growth of bifidobacteria and lactobacilli [25].

In the case of mango peels, which contained the highest ratio of SDF/IDF (40.1–41.1%), the production of SCFA was not as high as in orange or prickly pear peels. Nevertheless, during all fermentation times, significant differences (*p <* 0.05) were observed when MU and MP were compared. After 12 h of fermentation, an increase from 129.8 to 155.7 μmol/10 mg DF was achieved with the HPP treatment. Also, this treatment increased SCFA content after 24 h fermentation of MP when compared to the SCFA produced by the 24-h fermentation of the MU sample (from 183.5 to 204.6 μmol/10 mg DF).

According to Lebet et al. [26] and Hughes et al. [27], the amount of SDF is a determinant for the fermentability rate because it is consumed by the microbiota easily and totally. The SCFA profile in this case is affected by the chemical composition, especially in terms of the monosaccharide composition, linkage and molecular weight of the SDF. On the other hand, the fermentability and products of the IDF are not defined by its chemical composition; instead, they are defined by its physicochemical and structural properties. Thus, characteristics such as porosity or crystallinity define the susceptibility to microbiota digestion [27]. As explained above, the absence of significant shear forces that may change the structural properties of the peels during HPP would explain these results. Novel HPP technologies involving shear forces, such as microfluidization, have been investigated and developed to promote these structural changes that enhance the functionality of food ingredients. It has been shown that during these processes, a reduction of the particle size and an expansion due to the instant release of pressure are able to change the original structure of insoluble fibers by creating pores or cavities inside. Thus, these structural changes have an important effect on these physicochemical properties of these fibers, as has been demonstrated for wheat and corn [28,29,30].

The SCFA production might be related not only to the structure of the polysaccharide as stated by [2], but also to the ratio of SDF/TDF, the SDF-IDF content, the monosaccharide composition and physicochemical properties as stated by [13]. The differences in these features affected by the fruit peel matrixes yielded diverse SCFA profiles (Table 2). It can be noticed that during the 24 h of fermentation, in general for all the evaluated samples, the SCFA concentration followed the order acetate > propionate > butyrate and increased with fermentation time, as commonly found in the literature [25,31]. This is important because, under physiological conditions, SCFAs are absorbed when they are produced and, being slowly fermentable carbohydrates, they are preferably fermented when they reach the distal colon [24]. As in the present study, different AXs extracted from maize, rice and wheat developed different SCFA profiles [7].

Acetate proportions tended to decrease while butyrate tended to increase [24], because their metabolisms are closely related. Butyrate is relevant because of its relationship to the apoptosis of colon cancer cells [8]. Although FOS are considered one of the best butyrogenic sources, both orange peel samples generated similar amounts of butyrate after 24 h of fermentation (43.5 μmol/10 mg DF, Table 2). In fact, OU and OP showed a proportionally higher production of butyrate with a percentage ratio of 11.2–11.4%, as compared to 9.9% for FOS (Figure 5). On the other hand, it has been reported that orange peel pectins with different degrees of esterification yield diverse concentrations of butyrate after fermentation with the specific bacteria *Eubacterium rectale* [32].

For orange and prickly pear peels, the SCFA profile did not change with the HHP treatments. Napolitano et al. [25] reported similar SCFA productions and profiles despite significant changes in the SDF, IDF and TDF contents incurred due to HHP treatment. However, it is important to mention that even without significant changes in SCFA generation due to HHP, the treated DF stimulated the growth of hind gut bacteria differently [25,27]. In the specific case of mango peel, statistically significant increments were observed in the concentrations of acetate, propionate and butyrate when the untreated sample was compared with the HHP-treated sample, likely due to physicochemical changes after the HHP treatment as well as differences in sugar linkage proportions, as depicted in Figure 1.

## 3. Materials and Methods

### 3.1. Dietary Fiber Substrates

Orange (*Citrus sinensis* L. Osbeck) cv. Valencia (Hualahuises, N.L., Mexico (25.8807 N, 99.6235 W, altitude 371 meters above sea level), mango (*Mangifera indica L.* cv. Ataulfo) (Central de Abastos, Guadalupe, NL, Mexico) and prickly pear (*Opuntia ficus-indica* cv. Verde Villanueva) (La Flor de Villanueva, Acatzingo, Puebla, Mexico) peels were used for this study. Fruit peels used as DF substrates were obtained as described previously [13]. Briefly, fruit peels were manually removed, ground (VM0103, Vitamix, Cleveland, OH, USA), vacuum packaged (model EVD 4; Torrey, Monterrey, N.L., Mexico) in polyethylene bags and stored at 4 °C until use within 2 h. Then, untreated and HHP-treated peel samples were freeze-dried at −50 °C and 2.0 mbar (Labconco, Kansas City, MO, USA), hand-milled, sieved through a mesh number 40 (425 µm) and stored in desiccators containing P_2_O_5_ (25 °C) for at least 5 days before analysis.

Based on this study, HHP-treated fruit peels were chosen due to their characteristic composition profiles in terms of SDF contents, SDF/TDF ratios and/or functionalities. In addition, their processing conditions varied according to the type of fruit peel (Table 3).

### 3.2. Chemical Characterization

For all the samples, the composition analysis of neutral sugars (rhamnose, fucose, arabinose, xylose, mannose, galactose and glucose) and linkage positions of the monosaccharide residues were determined by their volatile alditol acetate (AA) derivatives and their partially methylated alditol acetate (PMAA) derivatives, respectively [33,34]. AA and PMAA derivatives in acetone were quantified by GC-MS (7890A and 5975 inert MSD with Triple-Axis detector, Agilent Technologies, Inc. Santa Clara, CA, USA) and separated in a HPLC capillary column (Supelco SP-2330, Sigma-Aldrich Inc., St. Louis, MO, USA) with the following conditions: injection volume 1 μL, injector temperature 240 °C, detector temperature 300 °C, helium as a carrier gas at 1.9 mL/min and a total run time of 30 min. Inositol was used as an internal standard and retention times were compared with neutral sugar standards. Composition was reported as the percentage of total carbohydrate content (dw). In the case of sugar linkages, their respective mass spectra were identified in the database of electron impact-mass spectra of partially methylated alditol acetates from the Complex Carbohydrate Research Center of the University of Georgia. The proportion of sugar linkages was reported as the proportion (%) of a specific sugar linkage with respect to the total identified sugar linkages.

### 3.3. Upper Gastrointestinal (GI) Digestion

To prepare samples for in vitro fermentation, an in vitro upper GI digestion was conducted according to [35] with slight modifications [7]. Briefly, 6 g of untreated and HHP-treated samples were mixed with 84 mL of phosphate buffer (20 mM, pH 6.9, 10 mM sodium chloride) and cooked for 20 min in boiling water to facilitate digestion. Enzymatic digestion was performed with salivary α-amylase (15 min), porcine pepsin (30 min) and porcine pancreatin (90 min) in a shaking water bath (150 rpm) at 37 °C. After digestion, samples were dialyzed (Spectra/Por1, MW cutoff 6–8 kDa, Spectrum Labs, Rancho Dominguez, USA) against distilled water for 26 h.

### 3.4. In Vitro Fecal Fermentation

Once the samples were digested, in vitro fermentation was carried out in an anaerobic chamber (85% N_2_, 5% CO_2_ and 10% H_2_) as previously described by [26] with brief modifications [3]. Carbonate-phosphate buffer (pH 6.8 ± 0.1) was autoclaved (121 °C, 20 min), then cysteine (0.1 g/mL) was added as a reducing agent (0.25 g/L buffer) and oxygen was removed by bubbling the buffer with carbon dioxide. Buffer was placed in the anaerobic chamber overnight. Samples were weighed (44 ± 0.5 mg) in test tubes considering 0, 6, 12, 24 h as fermentation time points, and placed into the anaerobic chamber. Then, 4 mL of carbonate-phosphate buffer were added to each tube.

Fecal samples were collected from three donors consuming their routine diets and who did not take antibiotics for at least 3 months. Fecal samples were placed in sterile plastic tubes, kept on ice during transferring, placed into the anaerobic chamber and used within 2 h of collection. Then, fecal samples were pooled and homogenized with carbonate-phosphate buffer (feces:buffer 1:10 (*w/v*)) and filtrated through four layers of cheesecloth, resulting in a fecal slurry. To each tube, fecal slurry (0.4 mL) was added for inoculation followed by closing with rubber stoppers, sealing and incubating in a water bath set at 37 °C and 150 rpm. Test tubes containing no substrate were used as blanks at each sampling period. Blanks were used to measure and correct any contribution from reagents but were not considered as part of the results. Also, FOS were used as a positive fast fermenting control. Triplicates were performed for each analysis. Protocols involving human stool collection and use were approved by the Institutional Review Board at Purdue University (IRB protocol #1510016635).

### 3.5. In Vitro Fermentation Products

#### 3.5.1. Total Gas Production and pH Changes Measurements

Graduated syringe displacement was used to measure gas production at each time point by passing the needle through the rubber stopper [8]. After these determinations, tubes were opened, and pH was measured with a potentiometer (Orion 3-Star, Thermo Fisher Scientific, Waltham, MA) previously calibrated with standard buffer solutions of pH 4.00, 7.00 and 10.01 (Orion^®^ pH buffers, Thermo Scientific Orion, Swedesboro, NJ, USA).

#### 3.5.2. SCFA Analysis

Aliquots (0.4 mL) from tubes were collected for SCFA analysis. Immediately after sampling, aliquots were mixed with 100 μL of the internal standard mixture (157.5 μL of 4-methylvaleric acid (nr 277827–5G, Sigma-Aldrich Inc., St. Louis, MO, USA), 1.47 mL of 85% phosphoric acid, 39 mg of copper sulfate pentahydrate) and the resulting blend was brought to 25 mL of final volume with purified water and 400 μL of copper sulphate solution (2.75 mg/mL) to halt fermentation. Samples were stored at −80 °C until analysis.

Defrosted samples were centrifuged at 3000× *g* for 10 min (Microfuge^®^ 20R, Beckman Coulter, Brea, CA) and 4 μL was prepared for injection into a gas chromatograph (model 5890 Series II, Hewlett Packard, Palo Alto, CA, USA) equipped with a fused silica capillary column (NukolTM, Supelco nr 40369-03A, Bellefonte, PA, USA) and a flame ionization detector (GC-FID 7890A, Agilent Technologies, Inc., Santa Clara, CA, USA). SCFA was assayed and identified as previously described [36] using acetate, propionate and butyrate relative to 4-methyl valeric acid as standards (Supelco, Bellefonte, PA, USA).

### 3.6. Statistical Analysis

Results were reported as means ± standard deviation. Determinations were carried out in triplicate. One-way analysis of variance (ANOVA) and Tukey tests were used for statistical analysis and for differences among means. All statistical evaluations were performed using the Minitab 17 Statistical Software (Minitab Inc., State College, PA, USA) with a significance level of α = 0.05.

## 4. Conclusions

There is an increased interest in specific DF functionalities, especially in terms of modulating microbiota present in the hind gut. The amounts needed to achieve specific biological effects and the delivery of such functional fibers are factors that play an important role in health. Several studies have been performed to evaluate the in vitro fermentation profile of purified compounds, but little information is available in terms of DF concentrates such as fruit peels and how they are affected by non-thermal processing. Orange, mango and prickly pear peels untreated and modified by HHP were tested in this study. The sugar profile was slightly changed after the HHP treatments. The major neutral sugar component was arabinose, followed by galactose and finally glucose. HHP treatments generally reduced the substituted sugars present in the three evaluated fruit peels. The production of SCFA after the in vitro fermentation of these fruit peels indicated that they have adequate fermentable fiber capacity. Among the peel sources, orange yielded the highest SCFA content during fermentation, followed by prickly pear and mango peels. Significant SCFA content increments and changes in the SCFA profiles were observed for HHP-treated mango peel after 24 h of fermentation. In general, HHP treatments affected the sugar linkage proportions of all the samples. It was observed that the HPP treatment changed the structure of the polysaccharides present by reducing the substituted sugars and these modifications changed the proportion and ratio of SDF and IDF. Although previous studies revealed that changes in the proportions of DF led to different functionalities in HHP-treated samples, a noticeable effect in the SCFA production was only observed in mango peels. Physical treatments applied to cereal fibers have shown that HPP treatments incorporating shear forces, for example throughout the use of microfluidization, may have a greater effect on the material and therefore change its physicochemical properties.

Further research is needed to study the effect of HHP in the growth of specific beneficial bacteria and their metagenomics to identify species present in the microbial community of the hind gut. Additionally, a correlation between functional properties such as viscosity promoted by the use of these fibers in food systems, SCFA production and the microbial species should also be established.

## Figures and Tables

**Figure 1 molecules-24-00697-f001:**
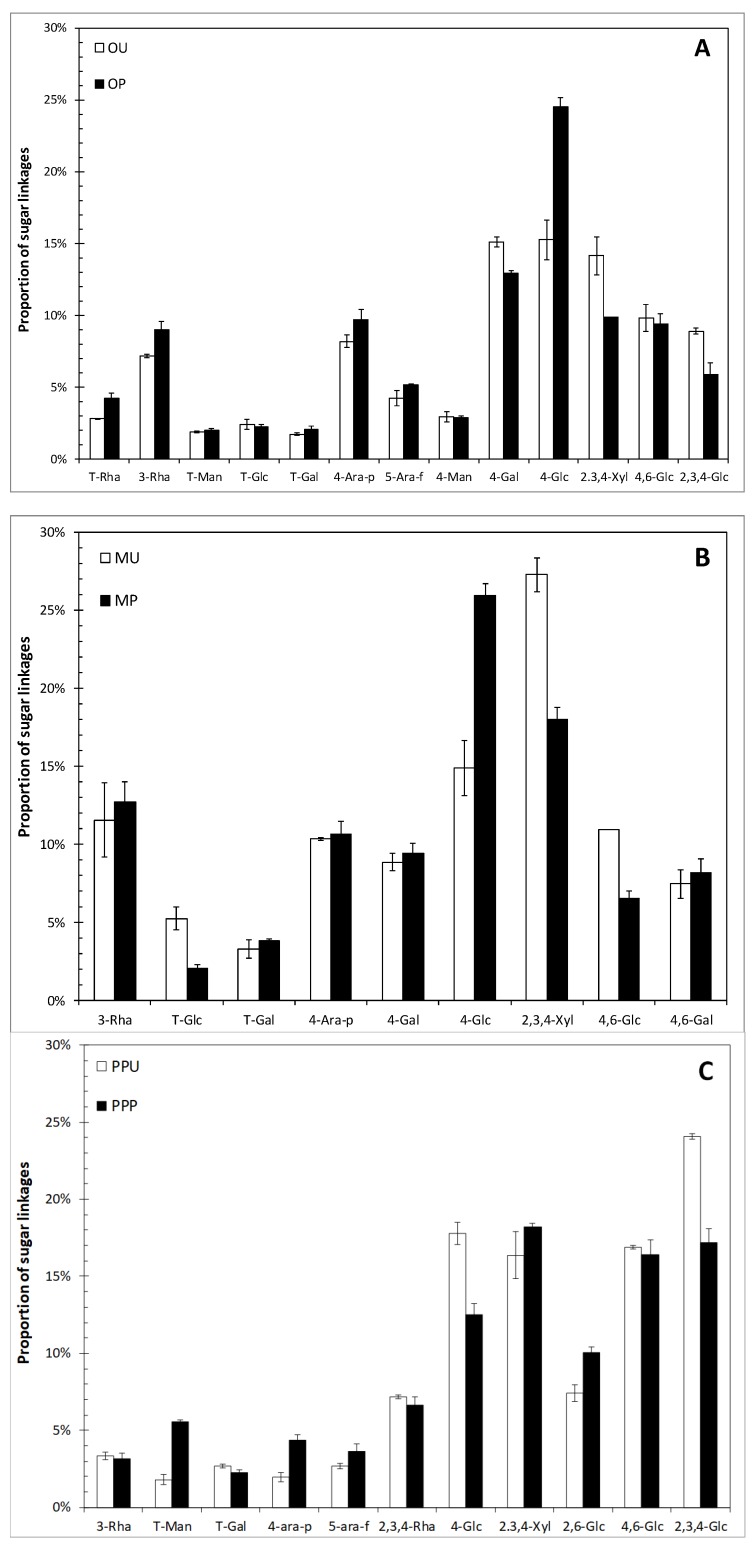
Sugar linkage proportions (%) of untreated and HHP-treated (**A**) orange, (**B**) mango and (**C**) prickly pear peels. Values are the means of triplicates and expressed in dw

**Figure 2 molecules-24-00697-f002:**
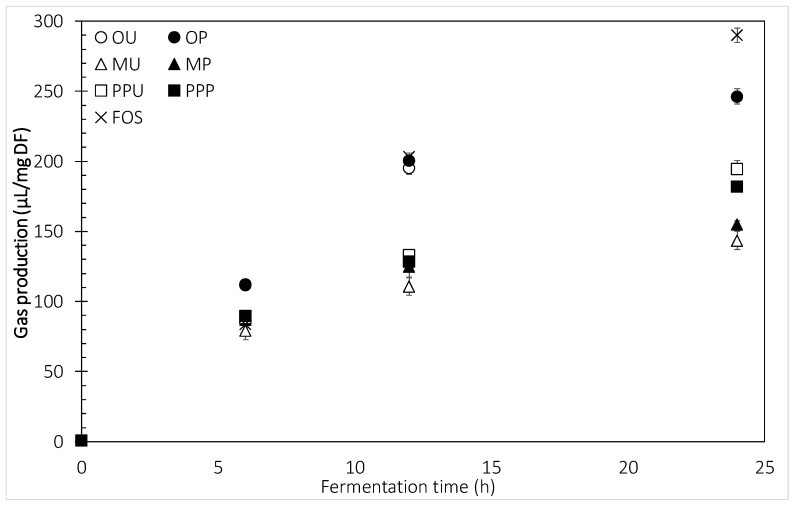
Gas produced during in vitro fermentation of untreated and HHP-treated fruit peels. Error bars show SD; some error bars are too small to see; *n* = 3. OU—orange untreated, OP—orange HHP-treated, MU—mango untreated, MP—mango HHP-treated, PPU—prickly pear untreated, PPP—prickly pear HHP-treated, FOS—fructo-oligosaccharides.

**Figure 3 molecules-24-00697-f003:**
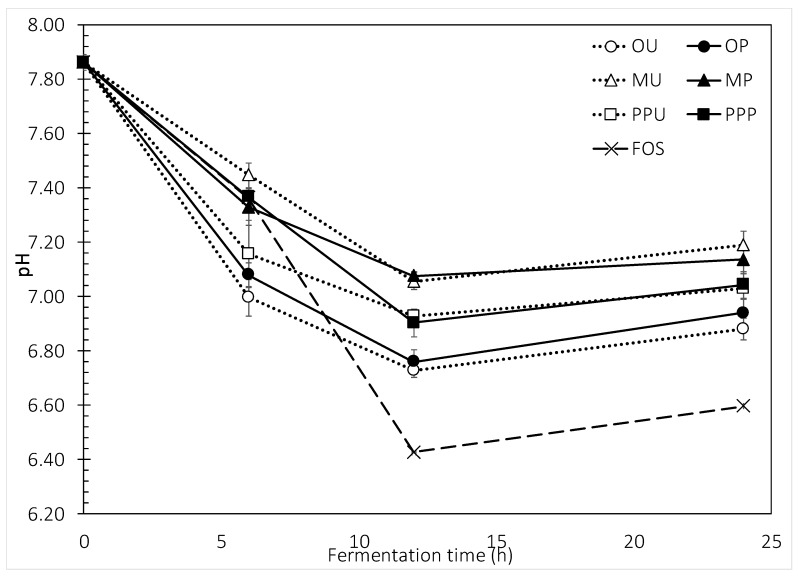
Changes in pH during in vitro fermentations of untreated and HHP-treated fruit peels. Error bars show SD; some error bars are too small to see; *n* = 3. OU—orange untreated, OP—orange HHP-treated, MU—mango untreated, MP—mango HHP-treated, PPU—prickly pear untreated, PPP—prickly pear HHP-treated, FOS—fructo-oligosaccharides.

**Figure 4 molecules-24-00697-f004:**
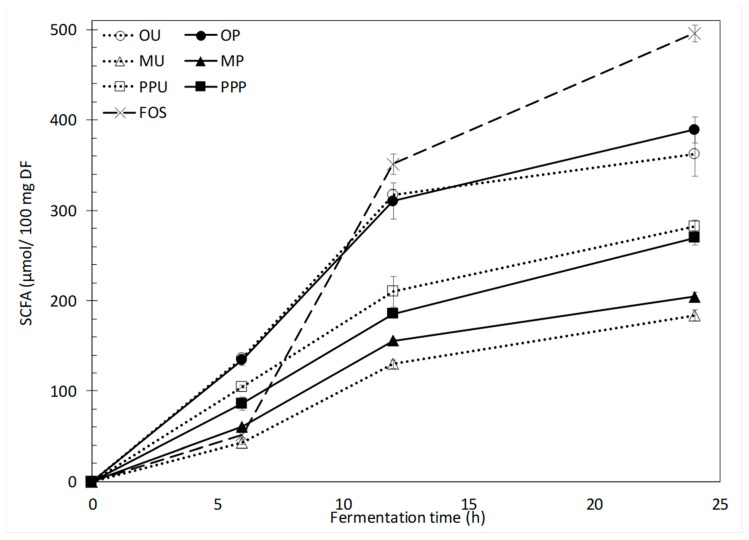
Short chain fatty acids (SCFA) produced during in vitro fermentations of untreated and HHP-treated fruit peels. Error bars show SD; some error bars are too small to see; *n* = 3. OU—orange untreated, OP—orange HHP-treated, MU—mango untreated, MP—mango HHP-treated, PPU—prickly pear untreated, PPP—prickly pear HHP-treated, FOS—fructo-oligosaccharides.

**Figure 5 molecules-24-00697-f005:**
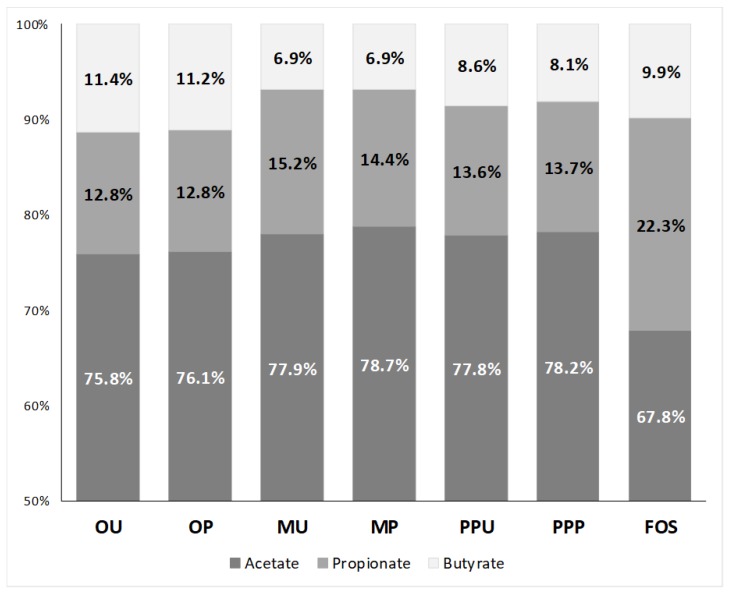
Acetate, propionate and butyrate percentage ratios after 24 h in vitro fecal fermentation of untreated and HHP-treated fruit peels compared to FOS. OU—orange untreated, OP—orange HHP-treated, MU—mango untreated, MP—mango HHP-treated, PPU—prickly pear untreated, PPP—prickly pear HHP-treated, FOS—fructo-oligosaccharides.

**Table 1 molecules-24-00697-t001:** Neutral sugars profile of orange, mango and prickly pear untreated and processed peels (% of total sugars).

	Orange		Mango		Prickly pear	
	*OU*	*OP*	*MU*	*MP*	*PPU*	*PPP*
**Rha**	5.71 ± 0.25*^a^*	6.22 ± 0.02*^a^*	2.01 ± 0.22*^b^*	2.63 ± 0.02*^a^*	4.22 ± 0.96*^b^*	7.61 ± 0.25*^a^*
**Fuc**	3.15 ± 0.04*^a^*	2.40 ± 0.09*^b^*	2.01 ± 0.16*^b^*	2.71 ± 0.10*^a^*	3.56 ± 0.38*^a^*	3.39 ± 0.19*^a^*
**Ara**	35.12 ± 1.51*^a^*	35.73 ± 0.47*^a^*	33.17 ± 1.61*^a^*	34.34 ± 1.94*^a^*	30.80 ± 1.09*^a^*	22.60 ± 0.58*^b^*
**Xyl**	7.26 ± 0.27*^a^*	7.48 ± 0.26*^a^*	5.76 ± 0.20*^b^*	6.60 ± 0.37*^a^*	12.85 ± 0.54*^b^*	14.24 ± 0.25*^a^*
**Man**	5.06 ± 0.03*^a^*	3.81 ± 0.25*^b^*	2.01 ± 0.09*^b^*	2.99 ± 0.37*^a^*	4.33 ± 0.04*^a^*	3.82 ± 0.23*^a^*
**Gal**	25.26 ± 0.84*^a^*	25.97 ± 0.63*^a^*	32.43 ± 0.83*^b^*	34.15 ± 1.35*^a^*	25.27 ± 2.01*^a^*	24.40 ± 1.74*^a^*
**Glu**	19.44 ± 0.71*^a^*	17.28 ± 0.41*^b^*	21.63 ± 1.75*^a^*	18.08 ± 0.17*^b^*	21.91 ± 1.61*^a^*	24.13 ± 0.98*^a^*

Mean value of four determinations ± SD; values per fruit, within the same row, followed by different letters are significantly different (*p <* 0.05). Rha—rhamnose, Fuc—fucose, Ara—arabinose, Xyl—xylose, Man—mannose, Gal—galactose, Glu—glucose. OU—orange untreated, OP—orange high hydrostatic pressure (HHP)-treated, MU—mango untreated, MP—mango HHP-treated, PPU—prickly pear untreated, PPP—prickly pear HHP-treated.

**Table 2 molecules-24-00697-t002:** Acetate, propionate and butyrate production (μmol/10 mg dietary fiber (DF)) during in vitro fecal fermentation with human fecal microbiota of untreated and HHP-treated orange, mango and prickly pear peels.

	Orange								Mango								Prickly Pear								FOS			
	OU				OP				MU				MP				PPU				PPP							
***Acetate***																												
6 h	119.3	±	2.7	*^a,1^*	117.2	±	5.0	*^a,1^*	43.3	±	2.5	*^a,1^*	54.2	±	3.9	*^b,1^*	86.6	±	4.6	*^a,1^*	72.9	±	4.7	*^a,1^*	45.5	±	4.4	*^1^*
12 h	243.4	±	4.5	*^a,2^*	239.7	±	14.8	*^a,2^*	101.6	±	1.9	*^a,2^*	121.9	±	2.6	*^b,2^*	168.1	±	12.6	*^a,2^*	149.4	±	2.6	*^a,2^*	273.8	±	8.6	*^2^*
24 h	274.7	±	18.2	*^a,3^*	296.3	±	10.4	*^a,3^*	143.0	±	5.5	*^a,3^*	161.0	±	3.7	*^b,3^*	219.4	±	5.3	*^a,3^*	211.0	±	7.9	*^a,3^*	336.5	±	4.0	*^3^*
***Propionate***																												
6 h	11.4	±	1.1	*^a,1^*	11.6	±	0.5	*^a,1^*	0.0	±	0.0	*^a,1^*	0.0	±	0.0	*^a,1^*	12.3	±	1.9	*^a,1^*	10.4	±	0.7	*^a,1^*	0.0	±	0.0	*^1^*
12 h	42.4	±	0.9	*^a,2^*	41.4	±	2.9	*^a,2^*	21.3	±	1.3	*^a,2^*	25.0	±	0.4	*^b,2^*	29.8	±	3.1	*^a,2^*	26.0	±	0.4	*^a,2^*	48.6	±	1.8	*^2^*
24 h	46.4	±	3.5	*^a,2^*	49.6	±	1.6	*^a,3^*	27.9	±	0.3	*^a,3^*	29.4	±	0.5	*^b,3^*	38.3	±	1.0	*^a,3^*	36.8	±	0.5	*^a,3^*	110.5	±	4.1	*^3^*
***Butyrate***																												
6 h	5.6	±	0.9	*^a,1^*	5.8	±	0.4	*^a,1^*	0.0	±	0.0	*^a,1^*	4.0	±	0.1	*^b,1^*	5.5	±	0.9	*^a,1^*	3.7	±	0.0	*^a,1^*	3.4	±	0.2	*^1^*
12 h	31.3	±	0.9	*^a,2^*	29.4	±	2.7	*^a,2^*	7.0	±	1.0	*^a,2^*	8.9	±	0.4	*^b,2^*	12.1	±	1.6	*^a,2^*	10.1	±	0.2	*^a,2^*	28.9	±	0.9	*^2^*
24 h	41.2	±	3.3	*^a,3^*	43.5	±	2.4	*^a,3^*	12.6	±	0.3	*^a,3^*	14.1	±	0.7	*^b,3^*	24.2	±	1.0	*^a,3^*	21.9	±	0.7	*^a,3^*	49.1	±	1.8	*^3^*

Mean value of three determinations ± SD; values per fruit (untreated and HHP-treated) within the same row followed by different letters are significantly different (*p <* 0.05); values per SCFA within the same column followed by different number are significantly different (*p <* 0.05). OU—orange untreated, OP—orange HHP-treated, MU—mango untreated, MP—mango HHP-treated, PPU—prickly pear untreated, PPP—prickly pear HHP-treated, FOS—fructo-oligosaccharides.

**Table 3 molecules-24-00697-t003:** Samples nomenclature and their processing conditions.

Fruit Peel	Treatment	Nomenclature
*Orange*	Untreated	OU
	600 MPa/55 °C/20 min	OP
*Mango*	Untreated	MU
	600 MPa/22 °C/10 min	MP
*Prickly pear*	Untreated	PPU
	600 MPa/55 °C/10 min	PPP

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
