# Peer review of "In Vitro Fecal Fermentation of High Pressure-Treated Fruit Peels Used as Dietary Fiber Sources"

_molecules, 2019, doi:10.3390/molecules24040697_

Round 1
Reviewer 1 Report
Please correct the manuscript according to the following observations:
1) Page 5 , line 137: “The physiological implications … relies in the selective consumption of DF by specific microbial groups. The above also changes the profile of colonic species…”
The statements of this paragraph should be explained, once the microbial profile was not identified. Or the microbial identification should be presented.
2) Line 347:
“… 4 μL in preparation for inyection into…”
Change the “y” to “j”
3) In “Materials and Methods, line 335”
About the total gas production, a graduated syringe was used to measure gas production and then the tubes were opened… So the results presented in Fig 2 are the volume of gas produced at 6h, 12h and 24h of fermentation. That is, the measured volume represents the quantity of gas produced at that time and not the amount accumulated along the fermentation.
Change the type of graph - should be presented by individual points not connected by lines.
4) In “Materials and Methods :
Describe the use of FOS at “In vitro fecal fermentation” .
5) The blanks were not commented or appear in the results. “Test tubes containing no substrate were used as blanks at each sampling period. “
Author Response
Attached you'll find our response to Reviewer 1.

Reviewer 2 Report
MS Number molecules-441716: In vitro fecal fermentation of high pressure treated 2 fruit peels used as dietary fibre sources
General comments
In 2017, Tejada-Ortigoza and collaborators published a paper on the dietary fiber profile of fruit peels and functionality modifications induced by high hydrostatic pressure (HHP) treatments. HPP was applied before to mango, orange, and prickly pear peels with the aim of modifying their dietary fiber and soluble dietary fiber contents. Based on this study, the team prepared the different substrates from the same fruits using the optimal processing conditions for each fruit (Table 3). In this study, the authors characterized the neutral sugar profile of untreated and treated HPP fruit peels. In vitro fecal fermentations were performed on untreated and treated fruit peels.
Overall, the study was well done. Are the results in Figure 1 really relevant? It is difficult to relate these results to those of faecal fermentations. Indeed, no analysis of the fecal microbiota by metataxonomy has been made, which does not allow determining the consumption of dietary fiber selection by specific microbial groups. This aspect is the main weakness of this study.
However, no major concern can be reported. The conclusion is clearly stated. The study is of interest and adequately meets the criteria of a good scientific paper.
No specific comments.
Author Response
Attached you'll find our response to Reviewer 2.

Reviewer 3 Report
The paper is interesting and in the scope of the Journal.
Some corrections must be performed previous to publication:
a) revise grammar and style.
b) errors in abreviations must be corrected (EXAMPLE: line 29. It is said "PPC" and it must be said "PPP").
c) line 93: the wide variation that is mentioned refers to variations between different raw materials or variations in relation to processing?
d) line 98 and 99: as it is not clearly explained in the EXPERIMENTAL SECTION the obtention of the samples of DF, it is not clear if the glucose detected is free glucose or glucose coming from polysaccharides. Please clarify this.
e) line 120: explain how do you evaluate non-branched glucose.
f) results for FOS are reported but in the material and method section, its use as a control or standard is not mentioned.
g) lines 350-351: Please re-write the paragraph. Which standards are you using?.
h) lines 354-355: Each experience was perfomed three times?. And the analytical determinations were perfomed 4 times?. Re-write the statistical analysis.
Author Response
Attached you'll find our response to Reviewer 3.
